# Partial Methane Oxidation in Fuel Cell-Type Reactors for Co-Generation of Energy and Chemicals: A Short Review

**Rodrigo F. B. de Souza** [1], **Daniel Z. Florio** [2], **Ermete Antolini** [3] and **Almir O. Neto** [1,*]

1   Centro de Celulas a Combustivel e Hidrogenio, Instituto de Pesquisas Energeticas e Nucleares, Av. Prof. Lineu Prestes, 2242 Cidade Universitária, São Paulo 05508-900, Brazil; souza.rfb@gmail.com

2   Centro de Engenharia, Modelagem e Ciências Sociais Aplicadas, Universidade Federal do ABC, Av. dos Estados, 5001, Santa Terezinha, Santo André 09210-580, Brazil; daniel.florio@ufabc.edu.br

3   Scuola di Scienzadei Materiali, Via 25 Aprile 22, Cogoleto, 16016 Genova, Italy; ermantol@libero.it

\*   Correspondence: aolivei@usp.br

**Abstract:** The conversion of methane into chemicals is of interest to achieve a decarbonized future. Fuel cells are electrochemical devices commonly used to obtain electrical energy but can be utilized either for chemicals' production or both energy and chemicals cogeneration. In this work, the partial oxidation of methane in fuel cells for electricity generation and valuable chemicals production at the same time is reviewed. For this purpose, we compile different types of methane-fed fuel cells, both low- and high-temperature fuel cells. Despite the fact that few studies have been conducted on this subject, promising results are driving the development of fuel cells that use methane as a fuel source for the cogeneration of power and valuable chemicals.

**Keywords:** fuel cell; methane to energy; methane to products; methane oxidation reaction; partial methane oxidation

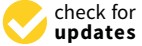



## 1. Introduction

Methane is the second most abundant greenhouse gas after $CO_2$, but is more potent in trapping heat, having a detrimental impact on the atmosphere [1]. On the other hand, this gas is the major component of natural gas, and has lower overall costs, which par excellence makes it a primary source for energy. There are large reserves of this gas, making it the main fraction of natural gas, coal bed gas, shale gas, and biogas, and it is also associated with oil extraction, where it is usually burned when in remote fields [2,3].

In a world that is geared towards reducing carbon footprint, methane has been identified as a possible source of transition energy, due to its low extraction and logistics costs, which can discourage the use of other fossil fuels [4,5]. However, even though the energy efficiency of burning liquefied methane is comparable to that of other liquid fuels, it is still used mainly for burning, generating $CO_2$ [4]. For this reason, the conversion of methane into chemicals is of interest to achieve a decarbonized future [2]. Direct conversion of methane, however, presents some challenges, such as breaking of the C–H bond, stronger than in any other hydrocarbon (434 kJ·mol$^{-1}$), and low polarizability, due to its regular tetrahedron conformation [6–11].

Methanol is an important industrial chemical, widely used in chemical synthesis, for the manufacture of many other chemicals [12–14]. It is also used in automotive antifreezes, in rocket fuels, and as a general solvent. Methanol is also a high-octane, clean-burning fuel that is a potentially important substitute for gasoline in automotive vehicles. Methanol is an alcohol present in nature in small amounts and obtained anthropologically by pyrolysis or destructive distillation of wood. Demand for methanol is expected to exceed 110 million tons per year (MTY) in the next few years and is expected to increase production capacity [15]. Methanol is commonly obtained from fossil fuels, but it can also be obtained from renewable energy resources, supporting, in this way, the development of energy

solutions with reduced greenhouse gas emissions. Besides their high-energy content, fossil fuels are hydrocarbon mixtures, that provide low-cost feedstock for the production of various platform chemicals [16], from fossil fuels, there is methane gas.

In most cases, the conversion process from methane to methanol is not completely selective, and, among different compounds, it is possible to find mainly formate. Other compounds with a higher carbon chain, such as ethane, ethanol, and propane, are also found, but less frequently and under more specific experimental conditions [17,18].

A classical process for conversion of methane into products involves the hydrocarbon and water decomposition to $CO + H_2$ (syngas) at high temperatures and pressures, an extremely energy-intensive process due to chemical inertness of the C–H bond. As such, the development of more efficient processes is an active field and a variety of methods have been explored [19–21]. Among other methods, these include homogeneous and heterogeneous catalysis, photocatalysis, biocatalysis, plasma technologies, and electrochemical processes [2,6,22–28].

All these processes can be conducted in mild conditions, such as the homogeneous catalytic process that consists of partial oxidation of methane by means of strong oxidizing species, $H_2SO_4$ [28,29] in oleum, and $H_2O_2$ [6,30,31], enhanced by metal ions, generating reactive oxygenated species.

In the bioconversion of methane in bacterial cells, Methane Monooxygenase (MMO) oxidizes methane to methanol using molecular oxygen and electron donors at room temperature and pressure [26,32,33]. This enzyme is biosynthesized by methane-oxidizing bacteria, which are microorganisms that utilize methane as a carbon and energy source [14]. The ability of MMO to cleave the C–H bond of methane under mild conditions is believed to originate from the generation of extremely reactive oxygen species at the catalytic site of MMO. Thus, the structure and properties of the catalytic site in MMO and the reactive oxygenated species involved in the catalysis have been investigated.

The photocatalytic process takes place by the ability of the catalyst to create electron–hole pairs, which generate free radicals [34–36], or reactive oxygenated species [6,37]. This process can be associated to hydrogen peroxide decomposition, enhancing conversion [34,37]. Shi and co-workers [37] synthesized and investigated a fluffy mesoporous graphitic carbon nitride (g-CN), for the partial oxidation of methane to methanol at a mild condition (35 °C) in the presence of $H_2O_2$ under the irradiation of simulated sunlight.

Li and co-workers [38] applied selective photocatalytic oxidation over graphitic carbon nitride-decorated tungsten bronze cesium nanocomposites to the conversion of low-concentration methane under mild conditions into methanol under light irradiation and at room temperature. The superoxide anion radical ($\bullet O_2^-$) first activated the methane and then the photogenerated electrons from nanocomposites inhibited the peroxidation and increased the generation of methanol [35,38].

Finally, electrochemical processes play an important role, as they act directly on the electrostatic interactions between an electron and atomic nucleus through the direct potential application on an atom or molecule, reducing energy dispersion in the environment [39]. This review presents a perspective of the oxidation of methane by electrochemical ways to obtain products with higher added value, preferably with co-generation of electricity. Discussing the application of fuel cells as reactors, the methane is consumed, its released electrons are used for electrical work, and products such as methanol are obtained. In a major plan, the potential of the application of crude natural gas ($CH_4 + CO_2$) and we discussed some of the most promising materials to be explored.

## 2. Electrochemical Methods for Methane Oxidation

Electrochemical processes can be carried out at both high and low temperatures, and present the advantage of transferring the energy directly from the hydrocarbon to the system, reducing the dissipation of energy in the form of thermal energy [39–43]. In addition, there is the possibility of selecting specific reaction products with the adjustment of work potential. For example, Ramos et al. [18] investigated methane oxidation over Pd/C using

a polymer electrolyte reactor—fuel-cell type at mild condition, to obtain methanol, formic acid, formaldehyde, ethane, ethanol, acetaldehyde, acetic acid and propane, reporting the onset potential for each product.

The electrochemical oxidation of methane can occur in two ways: the direct way, in which the hydrocarbon electron is transferred directly to the electrical system, and the indirect way, involving the electrogeneration of highly reactive species capable of reacting with $CH_4$ in the interface neighborhoods or/and in the bulk of the solution [44].

### 2.1. Faradaic Methane Oxidation

Faradaic methane oxidation is based on the adsorption of methane on the catalyst active site and on the direct electron exchange between the hydrocarbon and the electrode surface. Sustersic et al. [45] reported the adsorption of methane and its oxidation at various potentials on platinum in $H_2SO_4$ solution at 60 °C. The existence of two electrosorbed species, COH-type and CO-type, was highlighted through the current/potential characteristics recorded under different perturbation conditions.

Nandenha et al. [46] studied the Pt/C, Pt/C-ATO, Pd/C and Pd/C-ATO catalysts for methane oxidation: cyclic voltammograms of all electrocatalysts after adsorption of methane showed a current increase during the anodic scan, more pronounced for materials supported in C-ATO, related to the adsorption of methane and oxidation of the adsorbed species through the water activation. For all electrocatalysts, adsorbed CO or HCO intermediate species were observed by in-situ Attenuated Total Reflectance—Fourier Transformed Infrared (ATR-FTIR) experiments.

Boyd and co-workers [47] investigated the electrochemical conversion of methane to $CO_2$ on platinum under ambient conditions. Theoretical calculations suggested that methane is thermochemically activated on Pt surfaces ($CH_{4(g)} \rightarrow CH_3{}^* + H^*$) and that the stable methane-derived surface intermediate was CO*. They observed that catalysts with strong CO* binding and weak OH* binding energy are the most effective for high methane oxidation efficiency, but only at high potentials. These results are in line with those reported by Hahn et al. [48], which through FTIR studies, showed the existence of oxygenated carbonaceous species at potentials greater than 0.55 V over Pt, Au, Pd, Rh, and Ru. Arnarson and co-workers [49] identify a surface catalytic limitation of methane oxidation to methanol in terms of a compromise between selectivity and activity, as oxygen evolution is a competing reaction. By investigating two classes of materials, rutile oxides and two-dimensional transition metal nitrides and carbides, they found a linear relationship between the energy needed to activate methane, that is, to break the first C–H bond, and oxygen-binding energies on the surface. Based on a simple kinetic model, they concluded that to obtain sufficient activity oxygen has to bind weakly to the surface, but there is an upper limit to retaining selectivity.

Various authors, working on faradaic reactions, reported the importance of water activation, as one of the ways to transfer oxygen to the hydrocarbon molecule to obtain methanol [47,49]. Rocha [41] studied the methane oxidation in a batch reactor with $TiO_2/RuO_2/V_2O_5$ gas diffusion electrode and observed that the reaction occurs by a traditional mechanism in which the discharge of water generates hydroxyl radicals, which, in turn, chemically adsorb onto the oxide surface. This is in agreement with previous reports in the literature [9,50]; however, these radicals can not only react with adsorbed species. But also detach themselves from the bulk and react without electron transfer to the electrode. A scheme of the electrochemical reactor described by Rocha et al. [41] is shown in Figure 1, where is possible to see the methane inlet in an anodic gas diffusion electrode, in which methane is dispersed by the electrode and oxidized upon reaching the electrode interface with the electrode solution at controlled potential.

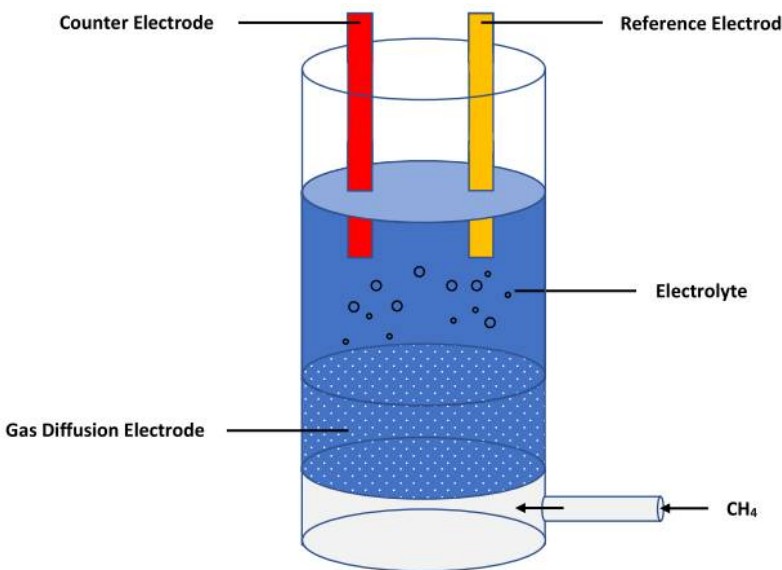

**Figure 1.** Scheme of the electrochemical reactor described by Rocha et al. [41].

*2.2. Non-Faradaic Methane Oxidation*

Methane can be activated by more reactive species originated by faradaic processes, that is, reactive oxygenated species (ROS), such as OH• [18,51,52], $O_2$• [53], and $SO_3$ [54,55], these species being capable to activating the C–H bond, without the need to absorb methane on the electrode.

Cui et al. [56] in a recent review, reported that the dissociation of $H_2O_2$ can generate OH• and these radicals produce $C_1$ compounds and $CH_3$• radicals. The $CH_3$• radicals can activate water and other hydrocarbon molecules. Cook et al. [51] reported a cathodic reaction to electrogenerate $H_2O_2$; then this specie is activated by Fenton reaction to OH• and finally this radical converts the methane into methanol. Neto's group [18,52,57] attributed the oxidation of methane at lower overpotential in other products, with co-generation of energy, to the activation of water and the presence of OH• radicals, arising from water activation.

Although ROS are the most commonly applied species for the partial oxidation of methane, other radicals can also be used. The use of a photo-electrochemical system to generate a chlorine radical, that reacts with methane to produce methanol was reported by Ogura [58]. $Cl^-$ is produced by an electrochemical reaction, and then this ion is turned into a radical under UV light. Cl• reacts with $CH_4$ generating methyl chloride, whose subsequent hydrolysis produces methanol.

Studies have demonstrated that the methane-oxygen cyclic operation was not necessary to obtain high selectivity [59,60]. The active oxygen species responsible for the catalytic conversion of methane into oxygenated products has to be clarified because the reaction could proceed either in the gas phase through a chain propagation mechanism [61–63] and/or on the catalyst surface [57,64–66].

Frese et al. [53] using an electrochemical cell to convert $O_2$ into $O_2$•$^-$ to obtain formaldehyde from methane, reported that there is a correlation between the current density and the selectivity, that is, increasing the current density increases the amount of ROS produced, and these species promote the production of more oxidized chemicals, such as CO and $CO_2$.

The propagation stage of radical reaction at mild conditions can promote the formation of non-oxygenated radicals, such as the methyl radical ($CH_3$•), and the detection of C2 and C3 compounds such as ethane and propane by Ramos et al. [18] is evidence of the formation of the methyl radical; these pathways are represented in Scheme 1.

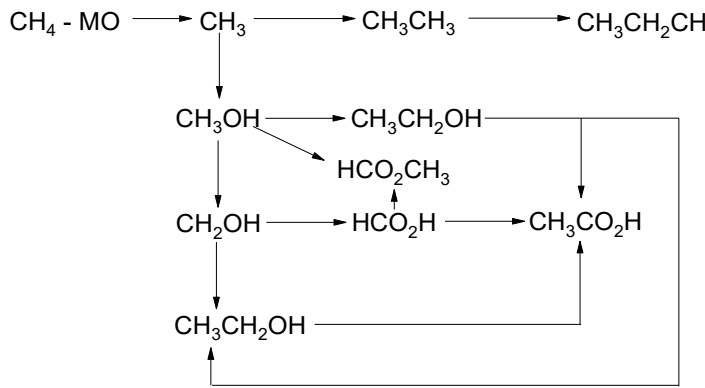

**Scheme 1.** Methane partial oxidation pathway proposed by Ramos et al. [18].

Methyl radical formation occurs by H abstraction from $CH_4$ by hydroxyl radicals. Gaseous methane first binds to the metallic sites [60], then hydrogen atoms are gradually abstracted by the adjacent surface M-O sites and yield M-OH. The reverse reaction of hydroxyl recombination is favored by $H_2O$ [6,36,60,67].

Methane activation chemistry and oxidative coupling of methane remain to date a subject of interest related to the initial methyl radical formation site and the role of surface-initiated gas-phase radical reactions [63,68–71].

## 3. Fuel Cells

Fuel cells are a tool commonly used to convert chemical energy into electrical energy, and are also used as a reactor for chemical transformations. Therefore, they can generate only energy (classic use), only valuable chemicals or both energy and valuable chemicals. While a previous review on methane-fed fuel cells was addressed to the production of only chemicals [10], this review deals with both energy and chemicals generation.

In a fuel cell, one reactant is oxidized at the anode and the other reduced at the cathode separated by a generally solid electrolyte. The oxidation of methane was tested in both compartments, for carrying out reactions by faradaic [72] and non-faradaic [73] pathways, as a showed in Figure 2. At first, the injection of methane into the anode was based on the assumption of a purely Faradaic reaction, however, the emergence of evidence that it could also occur by electrochemically initiated reactions for the activation of oxygenated species, it opens the possibility of cathodic injection of methane to convert it into products [73,74].

Vayenas et al. [75] compared the suitability of a cogeneration solid oxide fuel cells (SOFC) with respect to a chemical reactor of the same capacity giving the same product composition. Their model showed that such a relative suitability depends on the value of the dimensionless number, $\nu$, defined as

$$\nu = [24\alpha(KE - KH)tC/\alpha CC - CR] \tag{1}$$

where $KE$ is the unit electrical energy price (in $ kWh$^{-1}$), $KH$ the price per thermal unit produced in ($ kWh$^{-1}$), $tC$ the cogenerative SOFC useful lifetime (in days), $CC$ the cogenerative fuel cell capital investment per unit installed power (in $ kW$^{-1}$), $CR$ the chemical reactor capital investment per unit installed capacity (in $ day kW$^{-1}$) and $\alpha$ is a conversion factor given in the economic model. On these bases, a cogeneration fuel cell is more advantageous than a chemical reactor if $\nu > 1$.

Yuan et al. [76] compared thermal and electrochemical methods for methane partial oxidation. The thermal catalysis of methane partial oxidation has been explored over several decades using metals, metal complexes, metal oxides, and metal-exchanged zeolites bearing transition metal centers including Cu, Fe, V, Ir, Pd, Pt, Rh, etc. [77]. Unfortunately, these catalytic reactions usually require either strong oxidants, such as $N_2O$ and $H_2O_2$, or high thermal energy (when $O_2$ is used as the oxidant), to overcome the energy barrier for C–H bond activation or active site regeneration. The latter conditions often result in the

overoxidation of methanol to the thermodynamically more favorable product, $CO_2$. Thus, high selectivity to desirable products can only be achieved at low conversions resulting in a low yield [77].

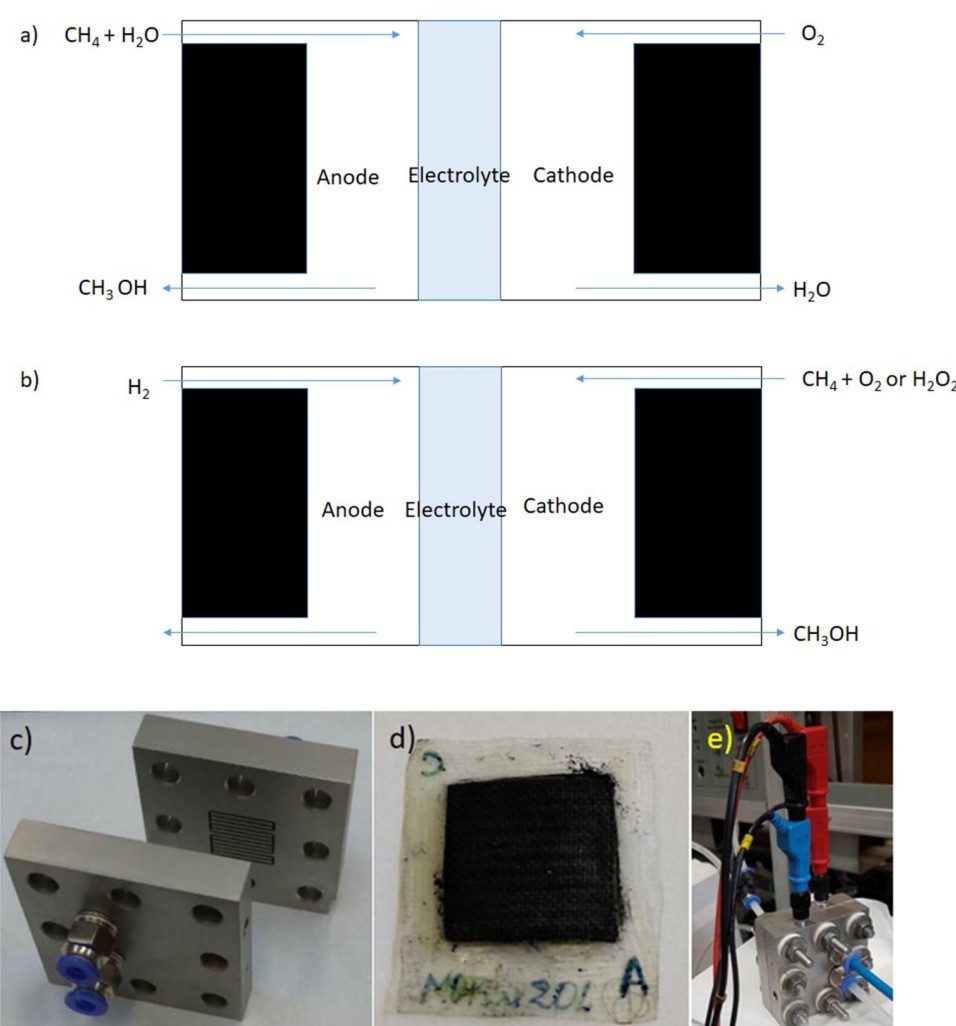

**Figure 2.** Fuel Cell Basic architecture: oxidation of methane (**a**) at the anode side and (**b**) at the cathode side. (**c**) reactor plates, (**d**) Membrane-electrode assemble, (**e**) reactor type fuel cell.

Thermal partial oxidation of methane over a catalyst is difficult to achieve because the Gibbs free energy for the complete combustion of methane to $CO_2$ and $H_2O$ is considerably more favorable than that for the partial combustion to methanol and formaldehyde. The high temperatures needed to activate methane create further complications because formaldehyde decomposes to CO and $H_2$ and methanol can undergo complete combustion. Consequently, high selectivity to formaldehyde and methanol is attainable only at low methane conversions [78]. Given the difficulties associated with the thermal partial oxidation of methane, interest has arisen in examining the possibility of oxidizing methane to methanol electrochemically at low temperatures (<150 °C) using water vapor as the oxidant. In the case of electrochemical oxidation, using water as the oxidant, the difference in Gibbs free energy for the oxidation of methane to methanol differs from that for the full oxidation to $CO_2$ by only 9 kJ/mol [78]. This small difference in energy suggests that it may be possible to achieve selectivity to methanol at appreciable conversion levels using an appropriate electrocatalyst. Summarizing, electrocatalysis has shown unique advantages to achieving an efficient and selective methane partial oxidation process by overcoming key limitations of the thermal systems.

At the beginning of the second half of the twentieth century, the use of hydrocarbons as fuel for fuel cells was considered, and the adsorption and oxidation characteristics of hydrocarbons were investigated [79–82]: it was reported that hydrocarbons adsorb slowly while undergoing dissociation on platinum electrodes in a narrow potential range around the final cathodic part of the double layer region and can be oxidized in an anodic potential sweep. Since the 1960s, with the observation that methane oxidation occurred only at high overpotentials, studies for non-protic electrolytes led to ceramic systems and subsequently to the use of hydrocarbons as fuel for SOFCs [83]. In the 21st century, with the improvement of proton-exchange fuel cells (PEMFCs), $CH_4$ started to be considered as fuel under mild conditions.

The performance of a methane-fueled proton exchange fuel cell was reported by Tyagi et al. [84], applying a Nafion membrane doped with phosphoric acid, obtaining an open circuit value of ~0.55 V using Pt catalysts. Nandenha et al. [46] using Pd/C as the anode catalyst obtained a power density of 0.5 mW cm$^{-2}$ in a PEMFC fueled by methane at 80 °C. In a later work, Nandenha et al. [85] obtained a power density of ca. 0.6 mW cm$^{-2}$, due to the effect of Zn addition to Pd, that enhances the water activation and the production of methanol and formate, which are more easily converted into energy than methane itself. Among different Pd-based catalysts, a power density of ca. 1 mW cm$^{-2}$ was obtained using PdAu as the anode catalyst, due to the methane interaction with the noble metals in the region of hydrogen adsorption; the formation of methanol and formate at water activation potentials was observed [86].

## 4. Fuel Cells for Cogeneration of Energy and Chemicals

Fuel cells can be applied for methane oxidation to obtain energy, added-value products, and both, and in all cases the differential is the material and operation conditions. As can be seen in Figure 2, methane oxidation can occur both at the anode or cathode side. Otsuka and Yamanaka [87] reported the partial oxidation of alkenes at low temperatures (<100 °C) using gaseous alkene–$O_2$ fuel cell systems and for hydroxylation of alkanes and aromatics applying $H_2$–$O_2$ fuel cell reactions, which control the reaction rate and selectivity by a variable resistor or potentiostat in the outer circuit. On this basis, Hibino's group [9,73] used a fuel cell as an electrolytic reactor to convert methane into methanol at different temperatures (50–250 °C), and attributed the high conversion to the production of $O_2$*$^-$ obtained from the activation of water over vanadium and tin oxides.

Spinner and Mustain [88], used a $CO_3^{2-}$ exchange polymer in a fuel cell-type reactor, to convert methane to $CH_3OH$, HCHO, CO, HCOO, and other low molecular weight oxygenate species at room temperature. The methane was activated electrochemically on a NiO-$ZrO_2$ electrocatalyst and the carbonate anions were used as an oxygen donor to facilitate methane activation.

Nandenha and co-workers [74] generated methanol through partial oxidation of methane at the cathode of a solid membrane reactor-PEMFC type ($H_2/H_2O_2$ + $CH_4$) at room temperature. $CH_4$ was injected into the cathode together with a $H_2O_2$ solution. They observed that the catalytic layer adsorbed methane and $H_2O_2$ in the active sites, generating OH• radicals, which reacted with the methane, giving rise to methanol formation. Methanol production was higher at the potential range of 600–400 mV.

Santos et al. [57] carried out the partial oxidation of methane for cogeneration of power and chemicals on Pt/C, Pd/C and Ni/C in an alkaline anion exchange membrane fuel cell (AAEMFC). FTIR data suggest that methane was converted into small organic molecules such as methanol and formate at different potentials for Pt/C, Pd/C, and Ni/C. The highest conversion efficiency was about 20% at 0.3 V using a Pt/C catalyst; the maximum conversion over Pd/C was 17.5% at 0.15 V, associated with the formation of a thin layer of PdO on the catalytic surface.

In later work, Neto's Group [89] investigated the effect of Ni content in PdNi/C catalysts used as anode materials in AAEMFCs on the fuel cell performance and methanol selectivity. They observed that the increase Ni content enhanced energy generation and the

formation of products more oxidized than methanol, probably due to the high ability of $NiO_x$ to activate water monitored by electrochemical experiments with Raman spectroscopy in situ.

For partial oxidation of methane by the electrochemical method, water activation is necessary [47,49], in which the hydroxyl radicals can be produced directly by the oxidation of water (Equation (2)) [90].

$$M + H_2O \rightarrow M(OH\bullet) + H^+ + e^- \tag{2}$$

This radical can be detected indirectly by the formation of hydrogen peroxide generated by the chemical equilibrium of this radical, according to Equation (3) [91]:

$$2\, OH\bullet\, M \rightarrow H_2O_2\, or\, HO_2{}^{-1}\, in\, alkaline\, medium \tag{3}$$

The detection of $H_2O_2$ electrochemically is carried out by rotation ring disk electrode (RRDE) experiments. Garcia et al. [52] used RRDE to give an indication of the most suitable percentage of Cu-complex, that should be used for the conversion of methane to methanol in an alkaline medium. Godoi and co-workers [92] used this methodology to evaluate $Pd_xCu_y/C$ electrocatalysts for methanol production from methane partial oxidation, and observed that the most active materials showed a content of $H_2O_2$ of about ~5–6%, similar to that reported by Garcia [52]; materials generating a $H_2O$ content higher than 6%, however, presented lower methanol production. For high $H_2O_2$ content, carbonate was detected in the products, inferring that the excess of ROS could lead to more oxidized products. The dependence on water activation alters the rate of reaction for methanol formation as shown in Figure 3.

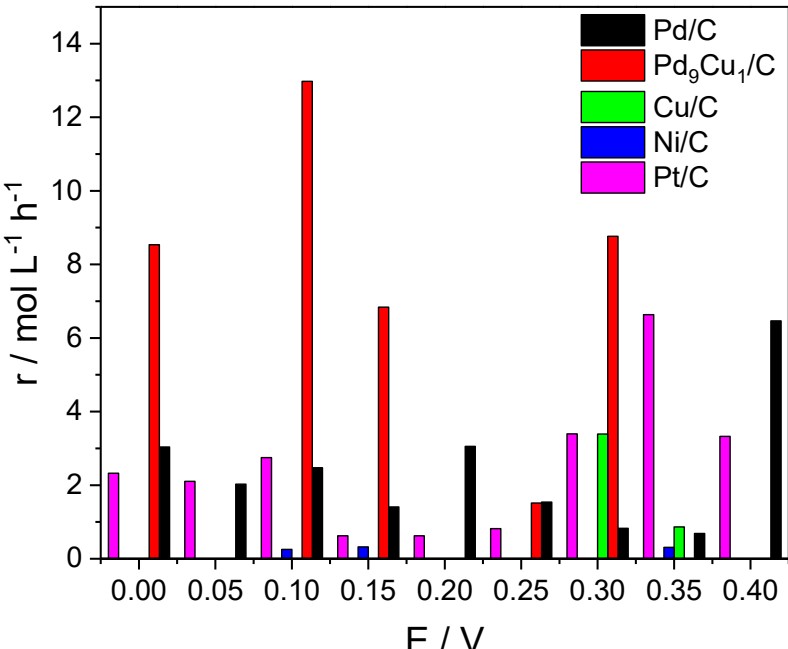

**Figure 3.** Rate reaction for methanol production in potential function of transition metals electrocatalyst, Data extracted by [57,89,92].

A collection of literature data on methane-fueled fuel cells including intrinsic parameters (catalysts and electrolytes), operational parameters (fuel composition and temperature), produced chemicals, and maximum power density (MPD) values is reported in Table 1.

**Table 1.** A collection of literature data on methane-fueled fuel cells including intrinsic parameters (catalysts and electrolyte), operational parameters (fuel composition and temperature), produced chemicals and maximum power density.

| Fuel Cell Type | Anode//Cathode | Methane Input | Eletrolyte | T/°C | Chemicals | MPD [a] /mW cm$^{-2}$ | Ref. |
|---|---|---|---|---|---|---|---|
| PEMFC [b] | Pd/C//Pt/C<br>Pd$_{(10\%)}$Au$_{(10\%)}$/C//Pt/C | Anode | Phosphoric acid doped Nafion membrane | 110 | - | 0.8<br>2.5 | [84] |
| PEMFC | Pd/C//Pt/C | Anode + H$_2$O$_{(g)}$ | Nafion | 80 | Formate | 0.55 | [46] |
| PEMFC | PdZn/C//Pt/C | Anode+ H$_2$O$_{(g)}$ | Nafion | 80 | Methanol, formate, formic acid | 0.6 | [85] |
| PEMFC | PdAu/C//Pt/C | Anode+ H$_2$O$_{(g)}$ | Nafion | 80 | Methanol, formate, formic acid | 1.0 | [86] |
| PEMFC | Pt/C//PdAu/C | Cathode+ H$_2$O or O$_2$ | Sn$_{0.9}$In$_{0.1}$P$_2$O$_7$ | 50–250 | Methanol and CO$_2$ | - | [73] |
| PEMFC | Pt/C//Pd–Au–Cu/C | Cathode + H$_2$O or O$_2$ | Sn$_{0.9}$In$_{0.1}$P$_2$O$_7$ | 50–450 | methanol | - | [93] |
| PEMFC | Pt/C//Pt/C | Anode + H$_2$O | Sn$_{0.9}$In$_{0.1}$P$_2$O$_7$ | 100–300 | Methanol | - | [9] |
| PEMFC | Pt/C//Pt/C | Cathode + H$_2$O$_2$ | Nafion | 80 | Methanol, formic acid and formaldehyde | 70 | [74] |
| AAEMFC [c] | Pt/C//Pt/C | Anode + KOH | KOH doped Nafion membrane | 25 | Methanol, formate | 0.3 | [57] |
| AAEMFC | PdNi/C//Pt/C | Anode + KOH | KOH doped Nafion membrane | 25 | Methanol, formate | 0.13 | [89] |
| AAEMFC | [6,6′-(2,2′-Bipyridine-6,6′-Diyl)bis(1,3,5-Triazine-2,4-Diamine)](Nitrato-O)Copper(II) Complex/C//Pt/C | Anode + KOH | KOH doped Nafion membrane | 25 | Methanol, formate | - | [52] |
| AAEMFC | Pt/Cu//Pt/C | Anode + KOH | KOH doped Nafion membrane | 25 | Methanol, formate | 0.12 | [92] |

[a] Maximum Power density. [b] proton exchange membrane fuel cell. [c] alkaline anion exchange membrane fuel cell.

## 5. High Temperature Fuel Cells

Solid oxide fuel cells (SOFCs), operating at high temperatures, thus minimizing polarization losses and the effect of impurities, are good candidates for the cogeneration of electricity and chemicals. Kiratzis et al. [72] used a SOFC for the chemical cogeneration of hydrogen cyanide, used in the synthesis of adiponitrile, in fumigation, as insecticide, in electroplating, metallurgy, and photography. This cell is formed by a tube of YSZ enclosed in a quartz tube, containing a porous Pt electrode as the cathode and a porous rhodium and platinum electrode as the anode, operating in the range 800–1000 °C, and using a mixture of methane and ammonia as the fuel, as the following anodic reaction:

$$CH_{4(g)} + NH_{3(g)} + 3O^{2-} \rightarrow HCN_{(g)} + 3H_2O_{(g)} + 6e^- \tag{4}$$

HCN selectivity depends on fuel composition, temperature, and current density. The cell selectivity to HCN could exceed 75%. The addition of oxygen to the fuel positively affects cell selectivity to HCN. For the currents tested, the only byproducts formed were CO and N$_2$ and a power density of ca. 0.01 Wcm$^{-2}$ was generated. Selective oxidation of methane in SOFC was studied for co-generation of C2 hydrocarbons and synthesis gas.

Oxidative coupling reactions of methane, such as the synthesis of ethylene and ethane, have aroused a lot of interest. Oxidation coupling of methane using a SOFC is an effective way to obtain C2 compounds and to generate energy at the same time [83,94–96]. The use of SOFC reactors for co-generation of C2 hydrocarbons and electrical power was first reported by Pujareand Sammells [94], in which a high selectivity to C2 hydrocarbons (>90%) was achieved but with low conversion.

Otsuka et al. [97] used a SOFC with yttria-stabilized zirconia (YSZ) as a solid electrolyte for the oxidative coupling of methane. Among various catalysts, the most active and selective was $BaCO_3$ (86% C2 selectivity) deposited on Au anode. The effects of temperature on the rates of products and C2 selectivity were investigated: the formation rate of C2 compounds and C2 selectivity showed maxima at 1073 K. The higher the pressure of methane, the better for the cogeneration of C2 hydrocarbons and electricity. An increase in the pressure of oxygen at the cathode also enhanced rate of C2 formation and current, but the C2 selectivity did not change appreciably.

However, the conversion of methane and C2 yields are still very low (<2%). Guo et al. [96] utilized a SOFC with 1 wt% $Sr/La_2O_3$-$Bi_2O_3$-Ag-YSZ membrane for the oxidative coupling of methane. An increase in the current generated was accompanied by a decrease in C2 selectivity and an increase in methane conversion. Methane conversion decreased, C2-selectivity increased and current generated decreased slightly with a rise in total flow rates. $CH_4$ conversion and the current generated increased with a rise in oxygen concentration. High methane concentrations are the more suitable conditions for the cogeneration of electrical energy and ethane and ethylene.

Tagawa et al. [83] used a SOFC for a selective oxidation reactor and for the conversion of the energy of oxidation into electric power. $La_{1.8}A_{l0}\cdot2O_3$ anode catalyst was effective not only to prevent the total oxidation of methane, but also to prevent carbon deposition. A thin thickness YSZ composite was used as the electrolyte in the SOFC reactor. A temperature higher than 1200 K the yield of valuable chemicals (CO and C2) was much higher than that of $CO_2$. The selectivity of $CO_2$ was always below 20%. Thus, this reactor can be regarded as a high selective oxidation reactor. The free energy change of the oxidation reaction has been directly converted into electric power with high efficiency.

The production of C2 hydrocarbons and electricity from oxidative coupling of methane in an SOFC reactor was simulated [95]. $La_{0.85}Sr_{0.15}MnO_3/8mol\%Y_2O_3$-$ZrO_2/La_{1.8}A_{l0.82}O_3$ (LSM/YSZ/LaAlO) were used as a cathode, electrolyte, and an anode, respectively. A plug flow reactor model (PFRM) was developed using kinetic parameters of the oxidative coupling of methane and the oxygen permeability through LSM/YSZ/LaAlO from the previous work [83]. Good agreements of power generation between experimental and simulation results were obtained. The effect of operating conditions was investigated. Methane conversion and C2 selectivity increase with increasing operating temperature. In this system, most C2 production is ethylene, which is more favored than ethane. Methane conversion decreases with increasing methane feed flow rate while C2 selectivity slightly increases. Higher methane feed concentrations on the anode give higher power. The reactor performance increases at higher pressures. The results suggest that this SOFC system is an excellent reactor for C2 production and power generation simultaneously.

Wiyaratn et al. [96] evaluated chemicals and power cogeneration in an $Au/LaSrMnO_3//$ $YSZ//LaSrMnO_3$ SOFC reactor. The e.m.f. and closed-circuit current increased to 0.86 V and 9.8 mA, (4.4 mW cm$^{-2}$) respectively at 850 °C. The e.m.f. was also close to the theoretical value of oxidation of methane. A 7% methane conversion and a 5.7% C2 hydrocarbons selectivity were observed when the air flow rate was 30 mL/min at 850 °C.

Some work has been addressed to syngas production and electrical power generation from methane-fed SOFCs [98–101]. Syngas can be used as feedstock for hydrocarbon and methanol production. In particular, methane partial oxidation expressed by the reaction

$$CH_4 + 1/2O_2 \rightarrow CO + 2H_2 \tag{5}$$

is attractive since it yields a synthesis gas with a molar ratio of 2. Synthesis gas with molar ratio of $H_2$ to CO of 2 is the most useful for the hydrocarbon and methanol synthesis. First, Yamada et al. [98] used a SOFC to obtain syngas by partial oxidation of methane. Co-doped $La_{0.9}Sr_{0.1}Ga_{0.8}Mg_{0.2}O_3$ was chosen as the electrolyte, since it was observed that doping small amount of Co was effective in increasing the oxide ion conductivity. The optimized composition for this electrolyte was $La_{0.9}Sr_{0.1}Ga_{0.8}Mg_{0.115}Co_{0.085}O_3$ considering the power density and the amounts of oxygen leakage. Although the thickness of the electrolyte with

the above composition was as thick as 0.5 mm, a maximum power density and yield of the synthesis gas were obtained at 242 mW cm$^{-2}$ and 16%, respectively, at 800 °C.

Then, Ishihara et al. [99] investigated the co-generation of syngas and electricity by using a single planar type SOFC. The electrolyte is a LaGaO$_3$-based oxides. La$_{0.6}$Sr$_{0.4}$CoO$_3$ and Ni were used as the cathode and the anode, respectively. A gaseous mixture of CH$_4$ and N$_2$ was employed as the fuel, while O$_2$ was used as the oxidant. The SOFC gave at 1000 °C a maximum power density of 526 mW cm$^{-2}$, which could be higher depending on doping, increasing the oxide ion conductivity. It was observed that doping LaGaO$_3$ based oxide with small amounts of Fe or Co was highly effective for increasing oxide ion conductivity. Thus, the use of Fe or Co doped LaGaO$_3$ as electrolytes in SOFCs for methane partial oxidation would increase the power density and the yield of synthesis gas. The yield of synthesis gas was around 20% in all the cases studied.

Sobyanin and Belyaev [102] investigated methane-to-syngas oxidation and power generation over Pt electrodes in an SOFC reactor. The experiment was performed at a constant ratio of methane and oxygen flows (CH$_4$/O$_2 \approx 1.8$), so that an increase in methane flow rate was equivalent to an increase in the current (oxygen flow). Syngas productivity increases linearly with the increase in CH$_4$ flow rate (or current), whereas electric power passes through at maximum. Zhang et al. [103] carried out the partial oxidation of methane to syngas at 700–800 °C in a SOFC with a Ni–SDC anode and an Sr- and Mg-doped lanthanum gallate in the composition La$_{0.9}$Sr$_{0.1}$Ga$_{0.8}$Mg$_{0.2}$O$_{3-\delta}$ (LSGM) electrolyte. They observed that the co-generated syngas at H$_2$/CO ratio of 1.4–2.0 varied with applied current densities, CH$_4$ flow rates and operating temperatures. The cell voltage at 100 mA cm$^{-2}$ and 800 °C was 0.90 V, i.e., about 90 mW cm$^{-2}$ power density could be obtained. The cell operating at 50 mA cm$^{-2}$ for 24 h almost showed no degradation of the cell performance.

Zhan et al. [102] used direct-methane SOFC (air,LSM-YSZ|YSZ|Ni-YSZ,CH$_4$) for power and syngas production simultaneously. Thermodynamic equilibrium analysis indicated that efficient methane conversion to syngas takes place for SOFC operating temperatures > 700 °C and O$^{2-}$/CH$_4$ ratios of ≈1. Fuel cells operated at $T \approx 750$ °C, $V \approx 0.4$ V, and O$^{2-}$/CH$_4 \approx 1.2$ yielded electrical power output of ∼700 mW cm$^{-2}$ and syngas production rates of ∼20 sccm cm$^{-2}$. Stable cell operation without cooking for >300 h was observed.

Pillai et al. [103] used direct-methane SOFCs to produce electricity and syngas. During initial operation at 750 °C, the cells produced 900 mW cm$^{-2}$ and ≈90% methane conversion to syngas at a rate of 30 sccm/cm$^2$. However, methane conversion decreased continuously over the first 30–40 h of operation, even though solid oxide fuel cells (SOFC) electrical performance was stable. An additional catalyst layer on the anode yielded a more stable methane conversion to syngas. Finally, the possibility of synthesis gas and power production from partial oxidation of CH$_4$ in SOFC was studied by Paloukis et al. through a numerical simulation [104]. By using a simple mathematical model, they showed the technological potentiality to cogenerate synthesis gas and electricity through the autothermal operation of an SOFC and its high performance with overall thermodynamic efficiency far above unity. This can be obtained by the SOFC operating under transient conditions by periodically reversing the flow through the monolithic reactor. Thus, the energy can be accumulated and trapped inside the reactor, maintaining the reactor at high temperatures (above the adiabatic temperature rise): in this way the electrochemical partial oxidation of methane into synthesis gas can proceed autothermally.

Brousas et al. [105] compared a regular SOFC plant (complete oxidation of methane to CO$_2$ and H$_2$O) with co-generation SOFC plants, producing ethylene and syngas. They observed that the rate of return for the regular fuel cell exceeded 25%, whereas that for the ethylene plant was about 21%, and that for the syngas plant was well behind at about 17%.

## 6. CH$_4$/CO$_2$ Mixtures

CO$_2$ and CH$_4$ largely contribute to the greenhouse effect. Conversely, conversion of CH$_4$ into syngas using CO$_2$ not only provides a useful feedstock for hydrocarbon, and

methanol synthesis but also contributes to the mitigation of greenhouse gases. The dry reforming of $CH_4$ with $CO_2$, an endothermic reaction ($CO_2 + CH_4 = 2CO + 2H_2$), is a viable method to convert $CH_4$ into syngas. Commonly, SOFCs directly convert the chemical energy of the fuels into electricity with $H_2O$ and $CO_2$ as the products, which is very promising in terms of energy efficiency yet leads to $CO_2$ emission in practice.

On the other hand, SOFCs are able to perform in situ $CO_2$–$CH_4$ reforming and $H_2$ selective electro-oxidation, enabling a sustainable path to produce electrical power and syngas from $CO_2$ $CH_4$ mixtures. Hua et al. [106] developed an innovative $CH_4$–$CO_2$ dry reforming process to co-produce electricity and CO-concentrated syngas, which benefits the selective oxidation of $H_2$ in high performance proton-conducting solid oxide fuel cells (H-SOFCs). An additional layer, consisting of a $Ni_{0.8}Co_{0.2}$–$La_{0.2}Ce_{0.8}O_{1.9}$ (NiCo–LDC) composite, was incorporated into the anode support, forming a layered SOFC configuration. The multiple-twinned bimetallic nanoparticles showed enhanced activity towards in situ methane dry reforming. Compared to conventional cell designs, this layered SOFC demonstrated drastically improved $CO_2$ resistance as well as internal reforming efficiency ($CO_2$ conversion reached 91.5% at 700 °C), and up to 100 h galvanostatic stability in a $CH_4$–$CO_2$ feedstream at 1 A cm$^{-2}$. More importantly, $H_2$ was effectively and exclusively converted by electrochemical oxidation, yielding no $CO_2$ but CO concentrated syngas in the anode effluent. The maximum power density exceeded 910 mW cm$^{-2}$ at 700 °C. The heat released by $H_2$ electrochemical oxidation fully compensated for that required by the extremely endothermic dry reforming reaction, making the entire process thermally self-sufficient. Moreover, the layered structure was beneficial in terms of decreasing coking and increasing $CO_2$ resistance of the SOFC in the mixed $CO_2$ and $CH_4$ feedstock. Summarizing, this H-SOFC showed outstanding electrochemical performances in $CH_4$–$CO_2$ feedstream, coproducing CO-enriched syngas with few $CO_2$ by-products.

Chen et al. [107] developed a multi-physical model to investigate the dry methane reforming (DMR) integrated SOFC running on $CH_4$ and $CO_2$ mixtures to obtain the co-generation of syngas/power. From the base case simulations at SOFC voltage of OCV and 0.7 V, they observed that the integration of proton conducting SOFC at 0.7 V could effectively enhance the syngas production than the OCV condition (4.8% improvement for $CO_2$ conversion and 21.6% for $CH_4$ conversion) and generate electricity (1.5 W) at the same time. The decrease of the voltage leads to higher electricity output, higher syngas production efficiency, and lower $H_2$:CO ratio of the product. However, the operating voltage is suggested to be higher than 0.5 V to prevent the undesired large temperature gradient. The inlet fuel flow rate showed a significant influence on the $CO_2$ and $CH_4$ conversion ratio.

The increase in flow rate gave rise to a low $H_2$ fraction, reducing the current density, and thus the temperature in DMR-SOFC. A collection of literature data on methane-fueled fuel cells is reported in Table 2.

**Table 2.** A collection of literature data on methane-fueled SOFC fuel cells including intrinsic parameters (catalysts and electrolyte), operational parameters (fuel composition and temperature), produced chemicals and maximum power density.

| Anode//Cathode | Methane Input | Eletrolyte | T/°C | Chemicals | MPD [a] /mW cm$^{-2}$ | Ref. |
|---|---|---|---|---|---|---|
| Pt or Rh//Pt/C | Anode + $NH_3$ | YSZ | 800–1000 | HCN | 10 | [72] |
| $Sm_2O_3$-$LaSrMnO_2$//$LaSrMnO_3$ | Anode | YSZ | 760 | C2 hydrocarbons | - | [108] |
| Sr/$La_2O_3$-$Bi_2O_3$-Ag//Ag | Anode | YSZ | 730 | C2 hydrocarbons | 6 | [94] |
| Au/$LaSrMnO_3$//$LaSrMnO_3$ | Anode | YSZ | 850 | C2 hydrocarbons | 4.4 | [96] |
| Ni//$La_{0.6}Sr_{0.4}CoO_3$ | Anode + $N_2$ | $LaGaO_3$ | 1000 | Syngas | 526 | [99] |
| Ni–SDC//$Sm_{0.6}Sr_{0.4}CoO_3$ | Anode | $La_{0.9}Sr_{0.1}Ga_{0.8}Mg_{0.2}O_{3-\delta}$ | 800 | Syngas | 90 | [101] |
| Ni-YSZ//$La_{0.8}Sr_{0.2}MnO_3$ | Anode | YSZ | 750 | Syngas | 700 | [102] |
| Ni//$NdBa_{0.75}Ca_{0.25}Co_2O_{5+d}$ | Anode + $CO_2$ | $BaZr_{0.1}Ce_{0.7}Y_{0.1}Yb_{0.1}O_{3-d}$ | 700 | CO-enriched syngas | 910 | [109] |

[a] Maximum Power density.

### 7. Materials for Methane Partial Oxidation Reaction

The activation of methane is influenced by the characteristics of the catalysts, such as their surface, structure, and composition. Transition metal oxides are the most commonly used in the conversion of methane-to-methanol [14,27].

The surface-catalyzed activation of C–H bonds of methane for its selective oxidative coupling into ethane and ethylene, or for its partial oxidation into methanol and formaldehyde is a subject of increasing interest. Many metal oxides have been claimed to be effective catalysts for these direct conversions of methane into functionalized chemicals.

Nickel is a frequently used metal catalyst for methane conversion due to its availability and low cost. There are different strategies for stabilizing Ni particles in catalytic reactions against coking and sintering. Some of them include modifications of Ni particles by the addition of a second metal, such as Co or Cu. Nematollahi and Neyts [110] by density functional theory (DFT) computations compared the reaction and activation energies of Ni and $NiN_4$ embedded graphene on the methane-to-methanol conversion near room temperature. Thermodynamically, conversion of methane to methanol is energetically favorable at ambient conditions. Santos and co-workers [57] attributed the conversion of methane to methanol on a NiO catalyst in a solid electrolyte reactor—alkaline fuel cell type to the low potential for water activation, behavior not observed for Pt or Pd. Their oxides were appointed as a site for methane adsorption and activation [88].

Platinum is one of the most efficient catalysts used in the oxidation of small organic molecules due to its excellent catalytic properties. When considering the electrochemical properties of Pt, it should be remembered that Pt surface structure alters the rate of adsorption and oxidation of the adsorbed molecules [47,111,112], i.e., terraces on Pt (100) can activate C–H bond at low energy [110]. Hsieh and Chen [113] studied Pt black for methane oxidation to $CO_2$ and reported the shift of onset potential for methane oxidation in function of pH.

Boyd and co-workers [47] found platinum to be the most active catalyst for the electrochemical oxidation of methane under ambient conditions. To understand the reaction mechanism and the factors that determine catalyst activity, they used experimental and theorical studies and found that the methane activation step is thermochemical (i.e., $CH_{4(g)} \rightarrow CH_3{*} + H{*}$). Nandenha et al. [74] used a Pt cathode catalyst in a solid electrolyte reactor—fuel cell type to activate $H_2O_2$ species and convert methane to methanol.

The catalytic activity of palladium can be enhanced by supporting it on carbon materials. Carbon materials such as carbon nanotubes (CNTs), activated carbon (AC), and reduced graphene oxide (rGO) were employed as catalyst support, and palladium-gold (Pd-Au) nanoparticles were used as active centers. The Pd-Au/CNTs catalyst showed outstanding methanol productivity and selectivity [112]. Compared with the PdAu/CNTs, Pd-Au supported on functionalized carbon nanotubes (CNTs-n) by a treatment of nitric acid enhances the methanol selectivity, but decreases the methanol productivity [71].

Serra-Maia and co-workers [27] observed that methane oxidation to methanol occurs in a significant excess of hydrogen peroxide by inhibiting the activity of AuPd nanocatalysts. Jin and co-workers [114] reported a catalyst system formed by Pd and Au strengthened by hydrophobically covered zeolite, to generate hydrogen peroxide for immediate reaction with methane at mild temperatures. Santos and co-workers attributed the Pd activity for methanol electrosynthesis to the formation of a thin layer of PdO on the catalytic surface [57].

Methane activation with in-situ $H_2O_2$ formation was successfully demonstrated in a microcapillary containing Au-Pd nanoparticles embedded in its silica-coated walls using a microchannel reactor. Direct methane activation with in situ generation of $H_2O_2$ promoted methanol formation [31,65]. The high affinity for oxygen of gold facilitates the breakdown of $H_2O_2$ in OH•, which would be the activator of the C–H breakage [71,115,116].

A bifunctional catalyst, with a carbophilic site and sites capable of activating water, as exemplified in Figure 4 can be used for maximize the conversion of the hydrocarbon into products, as a cited by Godoi [92].

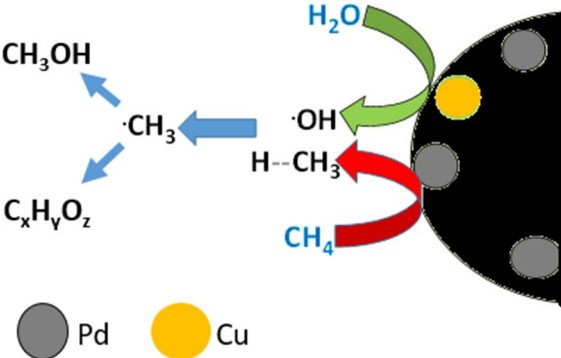

**Figure 4.** Partial methane oxidation with bifunctional PdCu/C electrocatalyst.

A screening of different metal oxides for the oxidative coupling of methane (OCM) by lattice oxygen, using first and second transition metal oxides (V, Cr, Co, Cu, Zr, and Mo) and some lanthanide oxides (La and Ce) was carried out by Hassan and co-workers [69]. The OCM activity was correlated with the surface amount of a low 1s binding energy (O1s, 526–527 eV) oxygen species on the catalyst, as well as its energy separation from the lattice oxygen.

Iron is extensively applied as a catalyst for the conversion of methane to formaldehyde [117], however, mild conditions make it more selective for less oxidized products such as methanol [118]. Yang et al. [6], in a study of visible-light-driven selective oxidation of methane to methanol on amorphous FeOOH coupled m-WO$_3$, reported the active role of Fe in the H$_2$O$_2$ decomposition mechanism. The same role was attributed to Fe by Zuo et al. [119] in a study about selective oxidation of methane with H$_2$O$_2$ over Fe-silicalite-1.

Copper is active for partial oxidation of methane into methanol, due to its high activity for water dissociation at 177–227 °C, a key process to C–H break [120], and the direct conversion increases in the presence of zeolites [14].

Sushkevich and co-workers [25] studied methane oxidation over copper-exchanged zeolites via chemical looping. Methoxy species and carbon monoxide are the primary reaction products formed directly from methane. Partial poisoning with water after the oxygen activation and before the methane reaction leads to the decrease of methanol yield, which is associated with the blockade of at least two copper atoms in the active site per one water molecule. In addition, they found that the poisoning was reversible and the increase in methane pressure increased CuMOR (copper mordenite) activity due to competitive adsorption on active sites. Alvarez and co-workers [19] also studied that the partial oxidation of methane to methanol over a Cu-MOR catalyst at 200 °C. They observed that at the optimum reaction conditions 52% of the adsorbed methane was converted to methanol. Metal organic frameworks (MOFs) are a class of hybrid crystalline materials, composed of metal ions/clusters linked together by organic ligands [121] which have a high surface area, microporosity that would act as reaction zones, limiting the residence time of the molecules of interest in the catalytic site, making the reaction more specific [122]. These materials have been applied for methane oxidation: Garcia et al. [52] used a [6,6′-(2,2′-Bipyridine-6,6′-Diyl)bis(1,3,5-Triazine-2,4-Diamine)](Nitrato-O)Coppe r(II) complex for the cogeneration of chemicals and power from methane.

Another material very active for water dissociation is ceria [120]; this oxide also acts as an oxygen buffer [123] in a thermoprocess. Lustemberg et al. [123] showed that a low-loaded Ni/CeO$_2$ (111) system activates CH$_4$ at 25 °C and then, with the help of water, which prevents sequential cleavage of C–H bonds and at 177 °C methanol was obtained with a conversion of about 35%.

Metal-exchanged zeolite has been reported as an efficient catalyst for partial oxidation of methane to methanol using only oxygen and water at low temperatures [61,124]. Kang and co-workers carried out selective oxidation of methane into methane oxygenates, includ-

ing methanol and formic acid, over Fe-zeolites and Pd/activated carbon in the presence of molecular hydrogen as a reducing agent using a liquid-phase with molecular oxygen [125].

Kang and co-workers [125] observed the partial oxidation of methane using Fe-zeolites prepared via an ion-exchange method to immobilize the homogeneous Fe catalysts into the zeolite matrix. Rocha et al. [41] applied the bismuth vanadate ($BiVO_4$) microcrystals for conversion of methane preferentially in methanol, explained that this selectivity is echoed in reports for the same reaction via photocatalyzed pathways.

## 8. Opportunities and Outlook

In fuel cell reactors, two main technical drawbacks have been identified: high prices and low stability. The catalyst material in fuel cell is the major obstacle to these applications. Up-to-date, although plenty of efforts in fuel cell catalysts have been focused on the synthesis of price-reasonable active catalysts with high stability, no true breakthrough has been reported yet. Thus, to develop catalysts with high activity by modifying chemical composition and physical characteristics (particle size, conductivity and dispersion of catalysts on the support), and stability (reduction of coke formation, tolerance to $CO_2$, CO, $SO_2$); and to reduce the cost are the important mission and challenge in fuel cell reactors for opening the opportunity in commercialization. In addition, the understanding of the reactions of fuel cell reactors including thermodynamics (entropy, cooling effect, hot spot) is required to develop the optimal condition for enhancing fuel cell activity.

In addition to being an efficient tool to generate power, fuel cells really show another very interesting application to produce valuable chemical as the main products together with the current delivered at the same time, so-called electro-chemical co-generation. There are many electro co-generation processes in different type of fuel cells, giving rise to a variety of valuable chemicals. Thus, electro co-generation has a major driven development towards commercialization. Overall, most industries should satisfy and should be able to apply electro co-generation in their own processes due to its show possibility for controlling the selectivity of product and % conversion with adequate current efficiency.

**Funding:** APC was sponsored by MDPI.

**Data Availability Statement:** The study did not report any data.

**Acknowledgments:** CAPES, CNPq (302709/2020-7), FAPESP (2017/11937-4) and CINE-SHELL (ANP) consented to the acknowledgement.

**Conflicts of Interest:** The authors declare no conflict of interest.

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
