# Peer review of "Partial Methane Oxidation in Fuel Cell-Type Reactors for Co-Generation of Energy and Chemicals: A Short Review"

_catalysts, doi:10.3390/catal12020217_

Round 1
Reviewer 1 Report
This manuscript reviews recent progress of using fuel cell-type reactors for cogeneration of chemicals and energy. Different types of electrochemical processes, methane-fed fuel cells, and catalysts for methane conversion are summarized, Future opportunities are also described. The topic is interesting and could attract a wide readership from researchers in the area of catalysts. Therefore, I recommend its publication after addressing the following issues.
- The main focus of this review is the electrochemical conversion, particularly via the fuel cell devices, of methanol to electricity and industrially useful products. However, in the introduction part, other techniques such as photocatalysis and biocatalysis are emphasized, The authors are advised to describe the electrochemical process, especially the fuel cell techniques in more detail.
- One more paragraph should be added in the introduction part for the description of the content and organization of this review article.
- In part 6, it seems that only the catalysts used at low and middle temperatures are described. How about the ones that is suitable for the application in high-tempertaurte SOFC, which is one the most promising techniques for methanol conversion and power cogeneration,.
- In page 15, second paragraph, the authors describe a photocatalytic activity and selectivity of BiVO4. The photocatalytic property is not related to the electrochemical conversion of methanol and electricity generation, and it is therefore better to delete this paragraph.
- In some parts, for example, from page 8 to 10 regarding the SOFC for cogeneration of energy and chemicals, although many examples are provided, descriptions of scientific insights are lacking for them. The authors are advised to describe the science behind these examples.
- There are some grammatical errors in the manuscript. It is advised to ask someone whose English is their native languages to check throughout the whole manuscript.
- The authors are advised to cite recently published relevant literatures such as Electrochim. Acta 2020, 342, 136118; ACS Appl. Energy Mater. 2020, 3, 3966−3977; Sol. Energy Mater Sol. Cells 2020, 207, 110349.
Author Response
Reviewer 1
- The main focus of this review is the electrochemical conversion, particularly via the fuel cell devices, of methanol to electricity and industrially useful products. However, in the introduction part, other techniques such as photocatalysis and biocatalysis are emphasized, The authors are advised to describe the electrochemical process, especially the fuel cell techniques in more detail. One more paragraph should be added in the introduction part for the description of the content and organization of this review article.
A: Was added a paragraph with electrochemical oxidation of methane and the description of the content and organization of this review in the end of introduction.
- In part 6, it seems that only the catalysts used at low and middle temperatures are described. How about the ones that is suitable for the application in high-tempertaurte SOFC, which is one the most promising techniques for methanol conversion and power cogeneration,.
The type of catalysts for high T SOFCs depends on the type of chemical cogenerated. For oxidative coupling of methane (OCM), an ideal catalyst has to hinder the total or partial oxidation of methane to CO2 and CO, respectively, and the secondary reaction of C2H4, allowing the OCM. A statistical analysis of the performance of OCM catalysts showed in the early 2010s found that most of the high performance catalysts were based on MgO or La2O3, with dopants for improving catalytic activity (Cl, Mn, W) and product selectivity (Na, Cs, Sr, Ba)]. SrO/La2O3 and Mn/Na2WO4/SiO2 are some of the best catalysts reported.
For syngas production, instead, the same catalysts for complete methane oxidation, generally Ni-based catalysts, have been used.
- In page 15, second paragraph, the authors describe a photocatalytic activity and selectivity of BiVO4. The photocatalytic property is not related to the electrochemical conversion of methanol and electricity generation, and it is therefore better to delete this paragraph.
A: the paragraph was removed and the electrochemically relevant about BiVO4 was rewrite.
- In some parts, for example, from page 8 to 10 regarding the SOFC for cogeneration of energy and chemicals, although many examples are provided, descriptions of scientific insights are lacking for them. The authors are advised to describe the science behind these examples.
Regarding methane coupling, the overall reaction to produce ethylene is:
2 CH4 + O2 à C2H4 + 2 H2O (..)
The half cell reactions are]:
2CH4 + 2O2- à C2H4 + 2 H2O + 4e- (..)
O2 + 4e- à 2O2- (..)
The reaction pathway is:
M+O- (catalyst) + CH4 à M+OH- + CH3● (..)
or:
CH4 + O2- à CH3● + OH- + e- (..’)
where ● denotes a radical species, followed by radical coupling in the gas phase:
2CH3● à C2H6 (..)
Ethylene is formed by gas phase thermal dehydrogenation of ethane radical:
C2H6 + OH- à C2H5● + H2O + 2e- (..)
C2H5● à C2H4 + H● (..)
Regarding syngas formation, oxygen ions are formed at the cathode, according to eq. (..):
O2 + 4e- à 2O2- (cathode reaction) (..)
Since the desired reaction is CH4 partial oxidation:
CH4 + O2- à CO + 2H2 +2e- (..)
total SOFC currents and methane flow rates are adjusted to yield an O2-/CH4 ratio ≈ 1. This is different than methane-fed SOFCs for the production of electricity by complete oxidizing methane oxidation, according to eq. (..:
CH4 +4O2- à CO2 + 2H2O + 8e- (..)
- There are some grammatical errors in the manuscript. It is advised to ask someone whose English is their native languages to check throughout the whole manuscript.
A: the grammar was reviewed
- The authors are advised to cite recently published relevant literatures such as Electrochim. Acta 2020, 342, 136118; ACS Appl. Energy Mater. 2020, 3, 3966−3977; Sol. Energy Mater Sol. Cells 2020, 207, 110349.
A: I believe the referee must have made a mistake when recommending the publications, because the one by electrochimica acta deals with the evolution of oxygen with nickel oxide and graphene in three-dimensional structures and the one by solar energy deals with the photo-electrochemical activation of water. If we look at the aspect of water activation, they could even be mentioned, but they do not bring any new fact that has not already been addressed. However we would very easily include them in a water activation review.
n this revised manuscript.
Reviewer 2 Report
This work aimed to review the partial oxidation of methane using different types of methane-fed fuel cells, both low temperature fuel cells and high temperature fuel cells, such as SOFCs for electricity generation and valuable chemicals production at the same time. This is an interesting topic, however, in my opinion, the work is not well organized, and major revisions will be suggested before its acceptation for publication.
- In introduction, some ways for conversion of methane into chemical products including homogenous and heterogeneous catalysis, photocatalysis and biocatalysis have been described, however, the application and charateristics of electrochemical processes based on fuel cells for methane conversion have not been illustrated. So, the topic and objectives are unclear.
- Fuel cells are a tool used to convert chemical energy into electrical energy, or used as a reactor for chemical transformations or both for energy and valuable chemicals, the related works and the corresponding schematic diagrams are suggested to provide for clearer description.
- In order to be consistent with the Abstract, low-temperature and high-temperature fuel cells for methane conversion aresuggested to be described by category.
- Materials For Methane Partial Oxidation Reaction are also suggested to be reviewed by category.
- Figure 3, Figure 4 and Figure 5 should be cited and detailed in the text.
Author Response
Reviewer 2
- In introduction, some ways for conversion of methane into chemical products including homogenous and heterogeneous catalysis, photocatalysis and biocatalysis have been described, however, the application and charateristics of electrochemical processes based on fuel cells for methane conversion have not been illustrated. So, the topic and objectives are unclear.
A: Was added a paragraph with electrochemical oxidation of methane and the description of the content and organization of this review in the end of introduction.
- Fuel cells are a tool used to convert chemical energy into electrical energy, or used as a reactor for chemical transformations or both for energy and valuable chemicals, the related works and the corresponding schematic diagrams are suggested to provide for clearer description.
A: This manuscript shows that there are three types of application commented on by the reviewer, for energy generating, for converting methane into products without obtaining energy, and for conversion of methane into added value products with co-generation of electricity. In all cases the schematic is similar, the difference being that for which type of application the materials applied and the operating potential. This topic was better clarified in the first paragraph of item 4 of the manuscript.
- In order to be consistent with the Abstract, low-temperature and high-temperature fuel cells for methane conversion are suggested to be described by category.
A: the high and low temperature part was separated in the text
- Materials For Methane Partial Oxidation Reaction are also suggested to be reviewed by category.
A: Separating materials into high and low temperature materials would not be very consistent, as the same material can operate in both conditions, giving the same effect and leading to the same products. there is an interchangeability between materials, and a more general view may be favorable to high and low temperature research.
- Figure 3, Figure 4 and Figure 5 should be cited and detailed in the text.
A: Was added
evised manuscript.
Reviewer 3 Report
Hello,
In general, the article is tackling a very important and interesting topic, reviewing and discussing partial oxidation of methane in fuel cell systems. However, some comments need to be considered for having the article better presented, and are summarized in the attached report.
Regards

Author Response
Reviewer 3
- Title: As the work reviews partial methane oxidation in the fuel cell, please indicate that the work is a review in the title.
A: Was added
- Some proofreading and grammar review are required, with some attention to subscripts.
A: Was done
- Please provide the full name for any abbreviations used such as PEMFC, AAEMFC, SOFC, FTIR, YSZ… when used for the first time.
A: was corrected
- References: a. Please do not lump references, and limit it to 3 references for each statement. Instead, specify the specific information obtained from the respective reference. b. Please specify the outcomes obtained in the work cited, for example, Shi et al. [38], Li et al. [39],….with more specific/quantitative outcomes as possible.
A: it is done every time it is really relevant, however, when it is a more general point of discussion and/or agreement between many highs we prefer to group
- More specific and relevant keywords should be used.
A: keywords was provide
6- Introduction: a. Please explain how methane utilization can help in achieving a decarbonized future. Is this statement through the life cycle footprint, or what does support such a claim. The further processing of methane should increase its carbon footprint, more specifically when converted into methanol, which is then used as fuel as indicated in the second paragraph on methanol. b. A better introductory paragraph for the electrochemical conversion of methane is needed.
- a. It is well known that methane has more than one source, including biomass (vegetable and animal), and using this source naturally reduces the need to extract fossil and/or mineral carbon. Soon a first reduction of the carbon footprint, because we are not taking sources that have already been fixed and throwing them into the atmosphere, but using what is already in the natural cycle. Because it is also a product released during the extraction of oil and coal, being a gas with much higher infrared light absorption than CO2 makes it an even worse greenhouse gas, and will be removed, and these are facts already scientifically accepted and incorporated into common sense. Thinking about converting hydrocarbons into products such as methanol, this product has industrial applications, which are not properly burned as fuel, although it can also be used for that, but much less than for other applications. methanol is also an interesting solvent, it can be converted more easily into other organic molecules among other applications as stated in the same paragraph cited by the reviewer. So logically, if we take advantage of a greenhouse gas and convert it into more stable products with other applications instead of releasing it into the atmosphere, we reduce the emission of atmospheric carbon and even remove part of the excess carbon from the natural cycle. b. the better introduction of electrochemical conversion was done.
- Electrochemical methods for methane oxidation: a. Figure 1 does not provide a unique representation or demonstration for the electrochemical system, being a very simple illustration, hence it is unnecessary to include. b. Scheme 1: still need some information, you may add the added or other reactant species on the arrow to get the product. For example, to move from CH3OH to CH3CH2OH, another reactant was added, which should be added on the reaction arrow.
- the figure 1 ilustred a concept not clear explained by Rocha, and the figure was created with the informations obtained with the authors. b. In addition to the methane and water vapor discussed, no other reactants are formed, the more complex products being the result of the formation of radicals and fragments of the same species formed, and which are more detailed in the article by Ramos et al.
- Fuel cells: You may provide a description and background information on the FCs types used for methane conversions, explaining their operation principles along with differences.
We don't find this aspect interesting, because there are some literature reviews that address this issue.
- Fuel cells for cogeneration of energy and chemicals: a. Maybe the section can be divided into subsections with each explaining the use of a certain FC type for methane conversion. Accordingly, table 1 can be subdivided into a respective table for each subsection, with more information represented in each table (hence the table should be moved to sec. 4). The tables reproduced for each section should have as well more performance information such as OCV, not only MPD, in addition to information on the chemical performance such as conversion, selectivities…. The tables should indicate feed stream composition (in case of a mixture, along with other operating conditions of pressure. b. The discussion in this section follows publication list nature rather than topic, I would highly recommend following the topic type discussion (instead of listing what each publication did and their respective outcome). Additionally, the authors should comment or discuss such obtained results and how it serves the discussion present in this manuscript. c. Figures 3 and 4 are not referred to in the text. d. I believe that Fig.3 showing FC for in situ Raman is not necessary, as there is nothing unique about the cell, nor about the application. e. The font used in the chemical reactions needs to be adjusted.
We revised the manuscript, adding some of the suggestions, however in our opinion the others could mischaracterize the main objectives of the present manuscript.
- Fuel cells for cogeneration of energy and chemicals: Maybe the section can be divided into subsections with each explaining the use of a certain FC type for methane conversion.
A: Was done the introduction and low temperature details and high temperature section
- Materials for methane partial oxidation reaction: I think the title should be catalyst material or catalysts.
A: materials was choose in detriment due there are materials that non interact with methane, however active species that interact with methane.
- Some discussion on comparing thermal and electrochemical methane conversion methods, in terms of scale, economics, and feasibility should be added.
A: the data are not clear in this regard, because in order to delve more deeply into this matter dissipated energy, there are almost no data provided by low temperature processes and economics and feasibility, cost in time and inputs are variables that change very easily with any small technological step.
- Maybe some better and more illustrative figures could be added to show some of the performance outputs for FC for methane conversion.
A: the figures was better described in this revised manuscript.
Round 2
Reviewer 2 Report
This manuscript could be suitable for publication after professional editing of English language.
Author Response
The manuscript was revised and the english language improved.
Reviewer 3 Report
Hello,
Thanks to the authors for considering the previously given comments. However, many comments still need to be considered for having the article better presented and can be summarized as attached.
Regards

Author Response
Response to Reviewers 3:
1- Title: I do not think the use of “About” in the title is appropriate, maybe “on” can be better.
A: Title rewrite
2- Still, some proofreading and grammar review is required, with some attention to subscripts.
A: Grammar was reviewed
3- Please provide the full name for any abbreviations used such as PEMFC, AAEMFC, SOFC, FTIR, when used for the first time, this is not done in the abstract, which is the first stop for readers.
A: In the abstract no more showed abbreviations
4- References: a. Please do not lump references, and limit it to 3 references for each statement. Instead, specify the specific information obtained from the respective reference. For those that you believe to be of common sense and wide agreement among many references, then 2 or 3 references are more than enough to support such claim, rather than 6 references used at the end of the second paragraph, or 9 references used at the end of the second paragraph in page#2. b. Please specify the outcomes obtained in the work cited, for example, Shi et al. [38], Li et al. [39] …. with more specific/quantitative outcomes as possible.
A: Some citations was divided and more details was provided, however, the authors did not find the limit of 3 references anywhere in the “instruction for author” nor in other reviews published by the same journal. For example: L. Goswami, A. Kushwaha, A. Singh, P. Saha, Y. Choi, M. Maharana, S.V. Patil, B.S. Kim, Nano-Biochar as a Sustainable Catalyst for Anaerobic Digestion: A Synergetic Closed-Loop Approach, Catalysts, 12 (2022).10.3390/catal12020186; C. Mendes-Felipe, A. Veloso-Fernández, J.L. Vilas-Vilela, L. Ruiz-Rubio, Hybrid Organic-Inorganic Membranes for Photocatalytic Water Remediation, Catalysts, 12 (2022).10.3390/catal12020180, both from the last edition. When more than 3 references are placed, it is to show that this fact is strongly supported by other works, even though they are points with some discussion.
5- Keywords: maybe “partial methane oxidation” will be better than “methane oxidation reaction” as a keyword.
A: I believe that “partial methane oxidation” should be added in the keywords, but it does not exclude the “methane oxidation reaction”.
6- Introduction:
A: there is no question
7- Electrochemical methods for methane oxidation: Figure 1 does not provide a unique representation or demonstration for the electrochemical system, being a very simple illustration, hence it is unnecessary to include. Additionally, the figure does not indicate the cathode and anode, what is the gas diffusion electrode: or is it meant to be a gas diffusion membrane?!
A: More information about this figure was added in page 3 last paragraph.
8- Fuel cells: You may provide a description and background information on the FCs types used for methane conversions, explaining their operation principles along with differences.
A: was added an explanation about the operation of fuel cells and some differences.
9- Fuel cells for cogeneration of energy and chemicals: a. Figure 2 does not have a reference. b. Maybe the section can be divided into subsections with each explaining the use of a certain FC type for methane conversion. Accordingly, table 1 can be subdivided into a respective table for each subsection, with more information represented in each table (hence the table should be moved to sec. (4). The tables reproduced for each section should have as well more performance information such as OCV, not only MPD, in addition to information on the chemical performance such as conversion, selectivities…. The tables should indicate feed stream composition (in case of a mixture, along with other operating conditions of pressure. The authors have developed only a section and named “hightemperature fuel cells” in response to this comment, which I see not exactly what has been requested, as there are no sections to discuss PEMFC and AAEMFC for example as lowtemperature FC. c. The discussion in this section follows publication list nature rather than topic, I would highly recommend following the topic type discussion (instead of listing what each publication did and their respective outcome). Additionally, the authors should comment or discuss such obtained results and how it serves the discussion present in this manuscript. d. I believe that Fig.3 showing FC for in situ Raman is not necessary, as there is nothing unique about the cell, nor about the application.
A: a) the figure 2 not have reference due is created by this manuscript with illustrated basic of methane inlet in fuel cell reactor. b) the table was broken in two tables, one for low temperature and other to high temperature. c) the manuscript was organized in an organized way to present what was and what is being produced in the area where there are still not enough works to show a predominant view on the rumors, but to show as many views as possible on the subject. so it looks a lot like a list of works. d) figure 3 was deleted
10- Some discussion on comparing thermal and electrochemical methane conversion methods, in terms of scale, economics, and feasibility should be added.
A: Some discussion on comparing thermal and electrochemical methane conversion methods, in terms of scale, economics, and feasibility was added.
11- Maybe some better and more illustrative figures could be added to show some of the performance outputs for FC for methane conversion.
A: we choose figures that we think are relevant to manuscript, but we accept suggestions from the editor and referees.
12- More constructive and structured conclusions should be provided in the opportunities and outlook section.
A: Was rewrite